# Sustainable Consumption Behaviour in Colombia: An Exploratory Analysis

**Alfredo Guzmán Rincón [1,*], Ruby Lorena Carrillo Barbosa [1], Ester Martín-Caro Álamo [1] and Belén Rodríguez-Cánovas [2]**

[1] School of Economic and Administrative Sciences, Corporación Universitaria de Asturias, Bogotá 110221, Colombia; lorena.carrillo@asturias.edu.co (R.L.C.B.); emartincaro@iep.edu.es (E.M.-C.Á.)

[2] Department of Business Organization and Marketing, Universidad Complutense de Madrid, 28003 Madrid, Spain; brcanovas@ucm.es

[*] Correspondence: alfredo.guzman@asturias.edu.co; Tel.: +57-321-254-03-63

**Abstract:** Sustainable consumption has positioned itself as an alternative for economic growth and social development because of its ability to deal with the future scarcity of natural resources and the prevention and mitigation of climate change, among other things. In this sense, the role of the consumer is preponderant, due to the fact their consumption behaviour has a direct effect on the environment; hence the importance of analysing their habits from different perspectives and social realities. Accordingly, the aim of this work is to explore the low-impact sustainable consumption behaviour in Colombia and the convergence and divergence of this type of consumer behaviour in the country. To achieve this, an exploratory, quantitative, and transversal methodology was used. The latter was based on a sample of 393 consumers to whom a self-report scale was applied in order to evaluate behaviours linked to quality of life, care for the environment, and resources for future generations. With the data collected, the following step to follow was to identify how consumers are grouped (hierarchical cluster analysis), what the differences are (single-factor ANOVA), the behaviours (descriptive statistics), as well as the relationship among them (Pearson correlation statistics). Results show that there are two consumer profiles with different levels of awareness of sustainable consumption behaviour. The principal outcome of the study was that Colombian consumers have embraced the behaviour of quality of life and resources for future generations; however, those consumers related to environmental care have been less involved, especially due to the influence of economic variables as such the cost of products and speculation in the prices of environmentally friendly products.

**Keywords:** sustainable development; sustainable consumption; behaviour; quality of life; environmental care; resources for future generations; Colombia

## 1. Introduction

Excessive consumption of goods and services combined with population growth has a negative impact on the environment, contributing to the acceleration of climate change [1–3]. In this sense, the Fifth Report of the Intergovernmental Panel on Climate Change recognizes that the impacts of this phenomenon on natural and human systems are varied, compared to the trajectories of 1.5 °C, 2 °C and 3 °C [4]. Nevertheless, in any of the scenarios, in which they establish the impact pathways and vulnerability derived from anthropogenic climate change, there will be a significant reduction in ecosystem services (e.g., changes in surface run-off, melting of the poles, extinction of corals and mangroves, etc.), and therefore a direct impact on human systems and the development of nations [4]. In this context, it has been established that faced with this vicious problem of consumption and population growth, it is necessary to break paradigms in the development of societies with the ultimate aim of preventing and mitigating the impacts of climate change [5,6].

Having said that, sustainable development and consumption emerge as an alternative for economic and social growth, which aims to protect the environment [7–9]. Thus, this type of consumption refers to the "The use of services and related products that respond to basic needs and provide a better quality of life, minimising the use of natural resources and toxic materials, as well as the emission of waste and pollutants throughout the life cycle of the service or product, so that the needs of future generations are not compromised" [10] as it was conceptualised in the Oslo Symposium led by the United Nations Economic and Social Council.

Therefore, since its conceptualisation in 1994, in different lectures, like The Johannesburg Plan of Implementation of 2002 and the United Nations Conference on Sustainable Development RIO+20, have been held with the aim of highlighting the importance and the willingness of nations to incorporate sustainable consumption into their development models. As a result, the former sought to have countries establish mechanisms and tools for their promotion in order to contribute to the prevention and mitigation of climate change [3]; and the latter ratified the Johannesburg Plan of implementation and the Ten-Year Plan of Programmes on Sustainable Consumption and Production Patterns.

Recently, and with the definition of the Sustainable Development Goals, the nations reaffirmed their commitment to establishing sustainable consumption and production patterns which seek to: 1. Implement the "Ten-Year Framework of Programmes on Sustainable Consumption and Production Patterns" in all countries; 2. By 2030 achieve sustainable management and efficient use of natural resources; 3. To reduce waste generation through prevention, reduction, recycling and reuse; 4. To encourage enterprises to adopt sustainable practices; 5. To massify the information and knowledge of sustainable development; 6. To develop countries' scientific and technological capacities to move towards sustainable consumption and production patterns, among others [11].

In this scenario, Colombia is no stranger to incorporating sustainable consumption into its development model. In this way, since Law 99, it has been established that "the process of economic and social development of the country will be guided by the universal principles and sustainable development contained in the Rio de Janeiro Declaration of June 1992 on Environment and Development" [12], and therefore, principle eight of this declaration was linked to this type of consumption as the articulating axis of sustainable development. As a result, in 2010 the National Policy on Sustainable Production and Consumption was defined, with the purpose of "creating a culture of sustainable production and consumption among public institutions, businesses and consumers" was established [13], p. 35. In the case of the latter, this policy expects them to include products with environmental quality in their selection criteria, which to a certain extent allows for innovation in more sustainable products and services [13].

Taking into consideration what has been said, the state recognized that it is not enough to incorporate sustainable consumption into its development model by assuming that consumers and their behaviour are the central axes, and therefore, they must assume their responsibilities in regards to the effects perceived in the environment [6,14]. Thus, in the case of Colombia, the Organisation for Economic Co-operation and Development estimated that consumption is the main source of $CO_2$ emissions, which are derived from land transport, industrialised manufacturing and agricultural activities. These emissions are equivalent to 0.4% of the world's total, however, due to population growth and unsustainable consumer practices, it is estimated that these emissions will increase by 50% by 2020 [15].

In this context, there is a demand from society, and in the particular case of Colombia, for the adoption of behaviours that promote the care of natural systems and the mitigation of the effects of climate change [6,16]. Hence, the need for a change of paradigm among consumers, moving from traditional consumption with a strong impact on the environment to one that is sustainable [17]. In this way, it can be observed that the study of sustainable consumption behaviour in Colombia is scarce if compared to research in other contexts such as the Anglo-Saxon or the European one, and the existing research has been

related to environmental education [18,19], sustainability and savings in public services in vulnerable populations [20], and trends for the development of products based on Green Marketing [21], so there is no holistic vision of these behaviours, or of the characterization of consumers based on them.

This paper aims to explore low-impact sustainable consumption behaviour in Colombia and the convergence and divergence of this type of consumer behaviour in the country by contributing to the enrichment of the literature on sustainable consumption behaviour from a holistic analysis as proposed by Quoquab et al. [17], who recognise that consumers do not necessarily have sustainable habits in all behaviour categories (quality of life, care of the environment and resources for future generations). This research premise has been characterised as being highly theoretical and the studies which analyse behaviour simultaneously are scarce [17,22]. As previously stated, the lack of knowledge about sustainable consumption behaviour in Colombia hinders the work of public policymakers, marketing specialists and other decision-makers in terms of sustainable development. For that reason, an analysis of this type of behaviour, which can provide new tools for the decision making, is required in order to legitimise the strategies, policies, regulations and decisions taken [19] by the Colombian government.

This article is divided into five sections: the first presents the literature review of sustainable consumption behaviour, with the main emphasis on low-impact behaviour; the second presents the methodology used in the development of this study; the third gives an account of the results; the fourth includes discussions of the results based on previous national and international research as well as the limitations of this research, and the fifth presents the conclusions of the study.

## 2. Literature Review

### 2.1. Conceptualising Sustainable Consumption Behaviour

Sustainable consumption behaviours are understood as habits in which consumers evaluate and consider the subsequent consequences of the use of goods and services for their quality of life, care of the environment, and resources for future generations [19]. These behaviours are categorised according to their level of impact and divided into high and low [23]. In the case of the former, they refer to those that, due to their nature, are not easily appropriated by consumers, because of the high economic costs that these products represent for them. Nevertheless, the implementation could bring great benefits to the environment, an example of which is the installation of solar panels, home water treatment plants, green walls, and eco-friendly architectural designs [24–26]. The former, or so-called low-impact, are those consumption behaviours that are easily appropriated, given that they do not involve a high budget, but, if practised in isolation by society, fail to mitigate or prevent the effects of consumption on the environment. Among this type of behaviour, the purchase of organic products, recycling, saving on public services, stand out. The latter is widely documented in the literature and have been evaluated from various perspectives such as sustainable and unsustainable practices [23]; the impact on the economy of the development of sustainable behaviours [27–30]; the self and social identity [31]; product development [32]; etc.

Due to the nature of this last type of behaviour, which can be incorporated by the general population as it does not require a large budget, it is feasible to appropriate it in countries with social inequality, such as Colombia [33]. Thus, sustainable consumption behaviours can be classified into three categories: quality of life, care for the environment, and resources for future generations.

### 2.1.1. Quality of Life

Regarding behaviours associated with quality of life, these refer to consumer habits that seek to avoid excessive purchases, as well as the measured use of goods and services

for the satisfaction of basic needs [17]. Therefore, the consumer gives priority to the purchase of goods or services which are strictly necessary and advantageous in comparison to the products the consumers already have and uses in different ways rather than those for which these products were initially designed [34]. In this respect, the literature shows that the adoption of such habits by consumers is often associated with the satisfaction of psychological needs, the ability to relate to others, economic autonomy and a sense of higher levels of well-being [34,35]. As a result, researchers who have analysed this type of behaviour have developed two approaches. The first concerns the eudaimonic perspective in which behaviours are associated with consumers' lifestyles, showing that low-income households can have the same level of functioning with second-hand products (reuse) as those in which only new products are purchased [36]. On their part, Max-Neef [37] and Blaitt [38] determined that the lifestyle developed in the ecovillages manages to satisfy the needs of subsistence, protection, affection, understanding, participation, leisure, creation, identity, and freedom, therefore, this style is not associated with low standards of quality of life. In this perspective, it has been shown that populations that adopt sustainable consumption behaviours of quality of life have a higher degree of psychological satisfaction, expressed in terms of self-acceptance, purpose in life, personal growth, autonomy, and positive relationships [39].

The second approach seeks to analyse specific consumption behaviours that may affect the quality of life of consumers. In line with this, Welsch and Kühling [40] identified that German consumers tend to have positive behaviours regarding the purchase of light bulbs and energy-saving appliances, as well as a preference for purchasing foods of plant origin over those of animal origin. According to Verhofstadt et al. [41], Belgians tend to buy fresh or seasonal products in order to reduce the impact of their consumption on the environment. However, they are not willing to change some behaviours such as the use of the vehicle, since they consider that their quality of life is affected, thus, in order to reduce their carbon footprint, they prefer to buy low emission vehicles. However, with regard to behaviour such as the recycling and reuse of products, the study carried out in the Vietnamese context by Nguyen et al. [42] determined that having positive attitudes, represented by the importance and ethical component of this type of behaviour, facilitates its appropriation by citizens, in addition to the positive influence exerted by the social circle and these types of behaviour, which makes it clear that consumers see recycling and reuse as a way of improving their quality of life and that of society in general.

Recent studies on this approach have incorporated new consumer practices such as online shopping, which show a significant improvement in the quality of life of consumers in many aspects, such as time savings, easy access to information, the development of more efficient communication channels and the development of new services. It is in the latter, where sustainable consumption behaviour is facilitated since it makes possible the reuse of products, exchange of properties, among others [43,44]. In the face of this type of behaviour, the present study is representative of this second approach. In Table 1, some examples of sustainable consumption habits related to the quality of life are presented as well as a synthesis of its conceptualisation.

### 2.1.2. Environmental Care

The behaviours related to the care of the environment refer to the habits of buying, using and disposing of goods and services that promote environmental care, minimising the use of toxic materials, waste, and pollutant emissions [17]. The environmental care behaviours are characterized by their appropriateness and adoption by consumers who tend to superimpose the welfare of ecosystems over personal tastes. These types of behaviours are generally associated with the intrinsic values of the products and their production chain.

Research shows that these behaviours are appropriate for consumers gradually over their lives and are therefore changeable and influenced by multiple variables [45–47]. In

this way, it is evident that access to information that accounts for the impact of consumption raises the concern of consumers with regard to environmental issues, leading them to change their consumption behaviour, an example of which is the purchase of organic products and the use of bicycles [48]. Other studies, such as the one carried out by Agrawal and Rahman [49], identified that the place of purchase, the product packaging, and other characteristics associated with it, act on behaviour since various emotions are generated that allow purchasing decisions to be changed and preferences for environmentally friendly products to be created. Similarly, Young et al. [50,51] and Aertsens et al. [52] highlighted the importance of the functional attributes of the products, which facilitate a change in previous habits to be generated, as they relate to superior quality, good taste, social well-being, etc.

Despite this, it is important to recognise that there are variables that prevent consumers from appropriating this type of behaviour [53–55]. Thus, the high economic costs of environmentally friendly products become the main barrier to their acquisition. As Mainieri et al. [56] and Hughner et al. [57] maintain, the economic factor limits the possibility of purchase by the general population, so only a small group can buy them. Among these categories are related technologies that use clean energy (e.g., electric cars) and in some countries, the purchase of organic food, as evidenced by Nguyen et al. [47], who recognised the preference of low- and middle-income consumers for conventional products by linking them to lower prices along with the preconception that organic products might have additional costs due to the greater time and resource consumption these green products need for the production which is represented by their unavailability at traditional shopping sites [58]. Another predominant variable is the social circle in which the consumer lives since consumer habits are influenced by this [59]; however, the studies carried out are not conclusive in this respect [58]. Some of the caring behaviours are exemplified in Table 1 as well as a synthesis of its conceptualization.

2.1.3. Resources for Future Generations

Resource-related behaviour for future generations means that consumers should avoid consuming too many natural resources, so as not to endanger their availability, while at the same time meeting their current needs [17]. In this sense, compared to the other two types of behaviour, this has not been of great interest to the academic community as it focuses mainly on the management of resources by companies or organisations [60,61], rather than on consumers' own behaviour regarding their habits in caring for resources for future generations. In this context, the study developed by Bulut et al. [62] identified that older generations practice behaviours related to energy efficiency, product reuse and environmentally conscious consumption. At the same time, studies practised in young people relate more conscious behaviours in which the materialistic consumption is eliminated [63,64]. Thus, they carry out key practices for sustainability, such as making reasonable use of cars [65], consumption of collaborative goods and services based on the economy of distribution [64], among others. Some examples of this type of behaviour are presented in Table 1 as well as a synthesis of its conceptualisation.

**Table 1.** Synthesis of sustainable consumption behaviours based on [17,37,42,53,56,61].

| Conceptualization | Behaviour | Example [1] |
|---|---|---|
| Quality of life refers to habits that aim to eliminate excessive purchases and the excessive use of goods and services without affecting the satisfaction of the consumer's basic needs. | Reduce product consumption and waste. | Food, drinks, water, electricity, etc. |
| | Recycle products. | Cardboard, paper, cans, etc. |
| | Reuse products. | Give clothes to charity, give second use to glass jars, etc. |
| | Buy products with biodegradable packaging. | straws, plates, bags, etc. |
| | Plan your purchases. | Market, vehicles, clothes, etc. |

| Conceptualization | Behaviour | Example [1] |
|---|---|---|
| Care for the environment refers to the habits of buying, using, and disposing of products that protect ecosystems. | Use ecological products. | Organic food, soaps, etc. |
| | Use biodegradable materials. | Bags, spoons, papers etc. |
| | Concern for the environment. | Product impact analysis. |
| | Pay extra resources for ecological products. | Buy products with green or eco-friendly stamps. |
| Resources for future generations represent the habits of consumers to avoid consuming too many resources that may be unavailable in the future. | Concern about the consumption of natural resources. | Save water, eliminate paper consumption, etc. |
| | Guarantee the availability of natural resources. | Minerals, water, fish, etc. |
| | Control impulse purchases. | Budget planning, market lists, etc. |

[1] the above are some examples of sustainable consumption behaviour, which does not mean that other types of behaviour are not adopted in the literature.

## 3. Materials and Methods

In order to fulfil the proposed objective, a study of an exploratory, quantitative and transversal nature was proposed. To this end, a non-probabilistic sample was taken, using the quota technique in a ratio of 1 to 3, from 393 Colombians based on the criteria proposed by Patton [66]. Here requirements for participation in the study were established as, being of legal age, residing in the country at the time of application of the instrument, and not being a foreigner. This was done in order to avoid distortions in behaviour by being located in a geographical space different from the national one. This type of sampling is helpful in exploratory studies as it enables the selection of those subjects who are accessible and acceptable for inclusion [67]. Additionally, national exploratory studies (e.g., the United Kingdom) have used similar sample sizes [51] to the one presented in this work. Table 2 summarizes the demographic characteristics of the sample.

**Table 2.** Synthesis of sustainable consumption behaviours.

| Characteristics | Results |
|---|---|
| Gender | Males: 186. |
| | Females: 207. |
| Age: | 18-25 years: 15%. |
| | 26-34 years: 33.6%. |
| | 35-49 years: 42.5%. |
| | 50-59 years: 8.1%. |
| | Over 60 years: 0.8%. |
| Educational level: | Undergraduates: 6.4%. |
| | Bachelor's degree: 37.2%. |
| | Professional and Advanced Degree: 35.4%. |
| | Master: 10.9%. |
| | PhDs: 0.2%. |
| Number of children: | No children: 39.4%. |
| | 1: 26.5%. |
| | 2: 22.9%. |
| | 3 or more: 11.2%. |
| Occupation: | Self-employed: 19.8%. |
| | Employed: 59.3%. |
| | Student: 13.5%. |
| | Retired: 5%. |
| | In charge of housework:1.5%. |

The self-report scale developed by Quoquab et al. [17] was used to evaluate low-impact sustainable consumption behaviour. This scale evaluates the three types of behaviour (quality of life, care for the environment, and resources for future generations). This scale is composed of 23 Likert-type items where 1 is equal to "totally disagree" and 5 to "totally agree". These items are associated with 12 behaviours as shown in Table 3. Data collection was carried out during the months of June and July 2020, using a specialised online survey platform.

**Table 3.** Self-assessment scale items Quoquab et al. [17].

| Code [1] | Item | Behaviour |
|---|---|---|
| QL1 | I always try hard to reduce miss-use of goods and services (e.g., I switch off light and fan when I am not in the room). | Reduce consumption. |
| QL2 | I recycle daily newspaper (e.g., use as pet's litter box, etc.). | Reuse products. |
| QL3 | I avoid overuse/consumption of goods and services (e.g., take print). only when needed) | Reduce consumption. |
| QL4 | I reuse paper to write on the other side. | Reuse products. |
| QL5 | While dining in restaurant, I order food(s) only the amount that I can eat in order to avoid wasting food. | Avoid product waste. |
| QL6 | I choose to buy product(s) with bi-odegradable container or Packaging. | Buy products with biode-gradable packaging |
| QL7 | I don't like to waste food or bever-age. | Avoid product waste. |
| QL8 | I recycle my old stuff in every pos-sible way (e.g., distribute old clothes among needy people. | Reuse products. |
| QL9 | I reuse shopping bag(s) every time go for shopping. | Reuse products. |
| QL10 | I plan carefully before I purchase a product of service. | Planning purchases. |
| CEW1 | I do care for the natural environ-ment. | Care for the environment. |
| CEW2 | I use eco-friendly products and services. | Use ecological products |
| CEW3 | I purchase and use products which are environmentally friendly. | Use ecological products |
| CEW4 | I often pay extra money to pur-chase environmentally friendly product (e.g., organic food). | Pay extra money to buy or-ganic products |
| CEW5 | I am concerned about the shortage of natural resources. | Care for the environment. |
| CEW6 | I prefer to use paper bag(s) since it is biodegradable. | Use biodegradable materi-als |
| CEW7 | I love our planet. | Care for the environment. |
| CFG1 | I always remember that my excess consumption can create hindrance for the future generation to meet up their basic needs. | Care about future genera-tions. |
| CFG2 | I care for the need fulfilment of the next generation. | Concern about the con-sumption of natural re-sources. |

| Code [1] | Item | Behaviour |
|---|---|---|
| CFG3 | I often think about future generations' quality of life. | Care about future generations. |
| CFG4 | I try to control my desire for excessive purchase for the sake of future generation. | Guarantee the availability of natural resources. |
| CFG5 | I am concerned about the future generation. | Care about future generations. |
| CFG6 | I try to minimise the excess consumption for the sake of preserving environmental resources for future generations. | Care about the consumption of natural resources. |

[1] Note: QL = Quality of life, CEW = Environmental care and CFG = Resources for the future generations.

With the data collected, an exploratory factor analysis (EFA) was carried out to determine whether sustainable consumption behaviours in Colombia were grouped into the three factors described by the self-reporting scale of Quoquab et al. [17]. To achieve this aim, the Kaiser-Meyer-Olkin (KMO) statistics, the Bartlett Sphericity Test (BTS), and the anti-imaging matrix were used to check whether the data were suitable for this type of analysis. Subsequently, using the parameters established by Cronbach, Godfred et al. and Comrey and Lee [68–70], the EFA was performed using the Varimax Rotation Main Factor Method, in addition to Cronbach's Alpha test to check the internal consistency of the items of each factor. For the latter, it was considered acceptable when the value of the statistic was between 0.60 and 0.80 and high when it was greater than 0.80 [68]. It is important to emphasize that the initial names of the factors of the instrument were respected so that the resulting ones were named according to the greatest number of items loaded into each of the factors.

With the confirmation of these factors, a hierarchical cluster binder analysis was applied to subdivide the individuals in the sample into groups with homogeneous characteristics in terms of sustainable consumption behaviour. In contrast to this type of analysis, Tan et al. [71] recognise that its use is appropriate when variables with descriptors are observed, as is the case with Likert-type scales. Based on the aforementioned information, the ward algorithm with Euclidean squared distance interval without value transformation was used. Subsequently, the differences between the k clusters identified by means of the single-factor ANOVA analysis were established, which was developed through the creation of three latent variables corresponding to the sum of the responses of each of the types of behaviour, namely, SUM_QL (summation of quality of life), SUM_CEW (summation of care for the environment) and SUM_CFG (summation of resources for future generations). Additionally, differences in behaviours were determined for each cluster, using the same test. Such a difference was considered statistically significant if the p-value was lower than 0.05.

Descriptive statistics and correlation analysis between the items were used to profile the consumers. In this way, it was defined that sustainable consumption behaviour was absent if the response on the Likert scale was 1, 2 and 3, otherwise, it was considered present. Regarding the correlational analysis, Pearson's statistic (r) was used, taking as a reference Cohen's [72] criteria for social sciences, where he considers low if r is between −0.29 to 0.29, moderate when r is between -0.3 to -0.49 or between 0.3 to 0.49 and high when r is higher than −0.5 or 0.5. The information was analysed using SPSS software, and the graphics corresponding to the triangular heat map were developed in Google Colab using Python programming language.

## 4. Results

For the EFA, the KMO statistic was equal to 0.94, therefore the variables were partially and strongly correlated. In the case of the BTS test, the value obtained from the Chi-square was 4965.18 with a *p*-value of 0.00, thus the study items were explained by the factors extracted in the present FAS. In a confirmatory way, the anti-image matrix of the variables showed strong partial correlations, since the values obtained are higher than 0.5 for each of them, as shown in Table 4. Based on what has been previously stated, it was determined through the sedimentation graph (Figure 1) that 50.19% of the cases are explained by three factors.

**Table 4.** Anti-image matrix correlation values [1].

| Code | Anti-Image Correlation | Code | Anti-Image Correlation |
|------|------------------------|------|------------------------|
| QL1 | 0.95 | CEW3 | 0.94 |
| QL2 | 0.94 | CEW4 | 0.90 |
| QL3 | 0.94 | CEW5 | 0.93 |
| QL4 | 0.94 | CEW6 | 0.95 |
| QL5 | 0.95 | CEW7 | 0.91 |
| QL6 | 0.92 | CFG1 | 0.93 |
| QL7 | 0.94 | CFG2 | 0.95 |
| QL8 | 0.93 | CFG3 | 0.94 |
| QL9 | 0.92 | CFG4 | 0.94 |
| QL10 | 0.90 | CFG5 | 0.92 |
| CEW1 | 0.95 | CFG6 | 0.96 |
| CEW2 | 0.91 | | |

Note: [1], sampling adequacy measures (MSA).

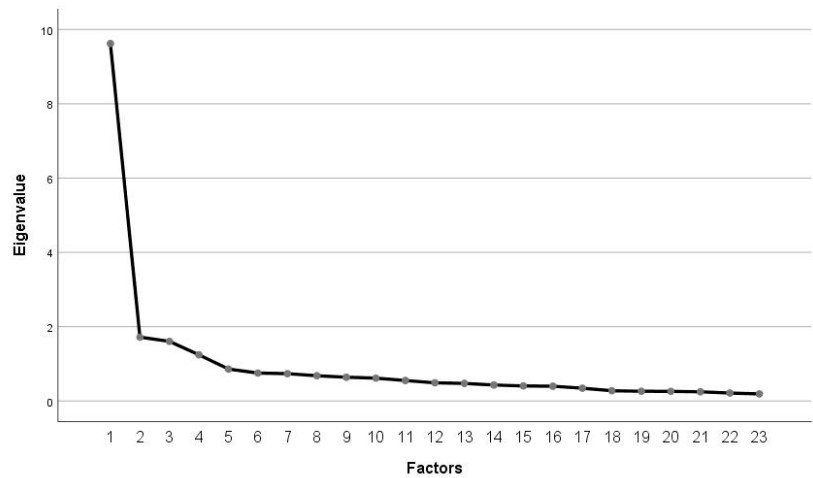

**Figure 1.** Sedimentation graph. Factors one, two and three, present values greater than one.

Thus, the EFA identified that 86.95% of the behaviours analysed on the scale were grouped according to the approach performed by de Quoquab et al. [17]. As a result, it was identified that in factor one (resources for future generations) CEW5, factor two (quality of life) CEW7, and factor 3 (care for the environment) QL6 were additionally grouped, as shown in Table 5. Taking as a base the conformation of these factors, for the case of resources for future generations the internal consistency of the items was 0.92, quality of life 0.84, and care for the environment 0.83, therefore, this consistency for the three factors was considered high from the parameters [65].

**Table 5.** Rotating factor matrix [1].

| Item | Factors | | |
|---|---|---|---|
| | 1 | 2 | 3 |
| QL1 | | 0.62 | |
| QL2 | | 0.47 | |
| QL3 | | 0.65 | |
| QL4 | | 0.62 | |
| QL5 | | 0.53 | |
| QL6 | | | 0.60 |
| QL7 | | 0.59 | |
| QL8 | | 0.46 | |
| QL9 | | 0.57 | |
| QL10 | | 0.34 | |
| CEW1 | | 0.45 | |
| CEW2 | | | 0.72 |
| CEW3 | | | 0.65 |
| CEW4 | | | 0.64 |
| CEW5 | 0.47 | | |
| CEW6 | | | 0.52 |
| CEW7 | | 0.50 | |
| CFG1 | 0.66 | | |
| CFG2 | 0.80 | | |
| CFG3 | 0.77 | | |
| CFG4 | 0.64 | | |
| CFG5 | 0.80 | | |
| CFG6 | 0.69 | | |

Note: [1], Extraction method: Factorisation of the main axis, Rotation method: Varimax with Kaiser normalisation and rotation has converged into 6 iterations.

After revising the results of the hierarchical cluster binder analysis, it was evident that the sample is divided into two conglomerates, taking as a reference the combination of distance-scaled cluster 15, thus, the first one was composed by 224 individuals and the second one by 169. The hierarchical blinder cluster dendrogram is represented in Figure 2.

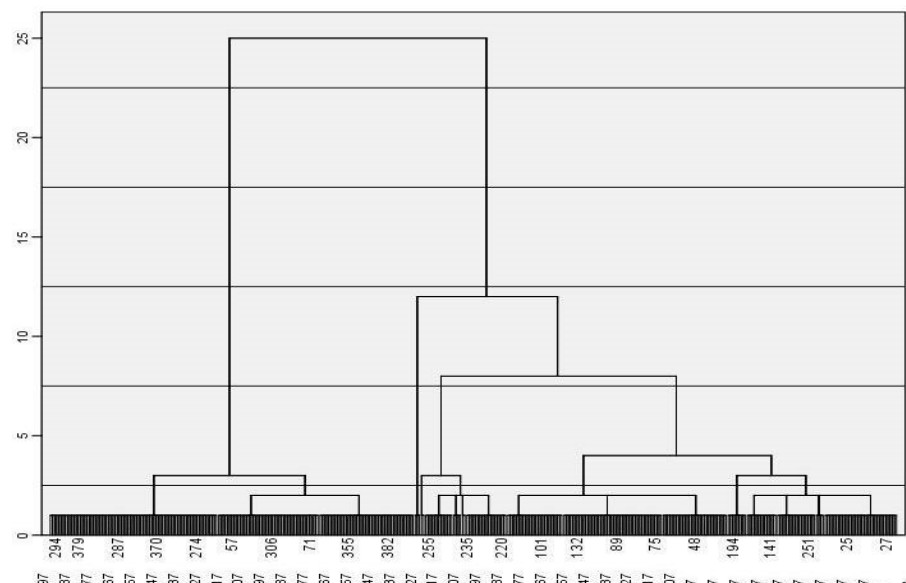

**Figure 2.** Hierarchical binder cluster dendrogram. Note: The X-axis represents the sample consumers, and the Y-axis represents the re-scaled distance cluster combination.

Regarding the statistically significant differences in the sustainable consumption behaviours of these clusters, it was identified that they have different scores in the latent variables. Therefore, for each of these results the sum of quality of life $F_{(1,392)}$ = 154.22, $p$ = 0.00, sum of care for the environment $F_{(1,392)}$= 233.94, $p$ = 0.00 and sum of resources for future generations $F_{(1,392)}$= 310.64, $p$ = 0.00. These differences are represented in Figure 3, whereby means of the scatter diagram it was shown that consumers grouped in cluster one, self-evaluated with lower scores in each of the sustainable consumption behaviours with regard to the latent variables.

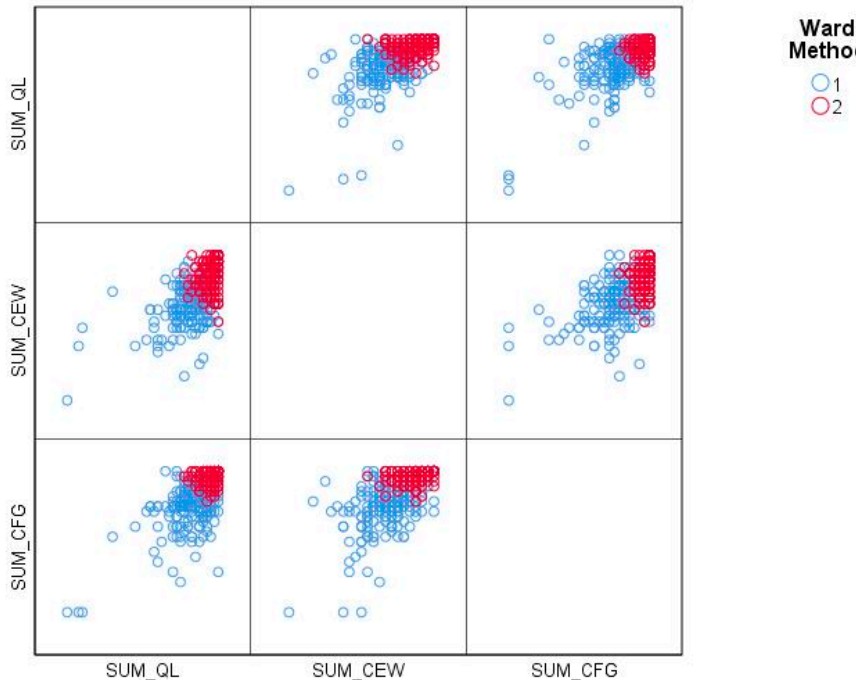

**Figure 3.** Cluster scatter diagram by type of sustainable consumption behaviour. 1 (cluster one) and 2 (cluster two).

However, with regard to the differences between the behaviours evaluated in the clusters, it was identified that for all of them there are statistically significant contrasts, which is represented in Table 6. It should be noted that the data are adjusted to a normal distribution, since the values of the Kolmogorov-Smirnov statistic, p-value, was higher than 0.05.

**Table 6.** One-factor ANOVA test per item.

| Item | Sum of Squares | Gl 1 | Quadratic average | F ² | *P*-Value |
|------|----------------|------|-------------------|-----|-----------|
| QL1 | 20.95 | 1 | 20.95 | 50.47 | 0.00 |
| QL2 | 70.41 | 1 | 70.41 | 69.93 | 0.00 |
| QL3 | 29.28 | 1 | 29.28 | 60.27 | 0.00 |
| QL4 | 25.62 | 1 | 25.62 | 51.77 | 0.00 |
| QL5 | 40.49 | 1 | 40.49 | 64.65 | 0.00 |
| QL6 | 81.25 | 1 | 81.25 | 118.72 | 0.00 |
| QL7 | 20.14 | 1 | 20.14 | 35.73 | 0.00 |
| QL8 | 24.51 | 1 | 24.51 | 47.68 | 0.00 |
| QL9 | 27.25 | 1 | 27.25 | 48.47 | 0.00 |

| Item | Sum of Squares | Gl [1] | Quadratic average | F [2] | *P*-Value |
|------|---------------|------|-------------------|-----|---------|
| QL10 | 36.45 | 1 | 36.45 | 61.93 | 0.00 |
| CEW1 | 32.86 | 1 | 32.86 | 77.95 | 0.00 |
| CEW2 | 73.96 | 1 | 73.96 | 144.17 | 0.00 |
| CEW3 | 67.05 | 1 | 67.05 | 133.99 | 0.00 |
| CEW4 | 71.80 | 1 | 71.80 | 78.89 | 0.00 |
| CEW5 | 30.50 | 1 | 30.50 | 76.63 | 0.00 |
| CEW6 | 52.80 | 1 | 52.80 | 76.32 | 0.00 |
| CEW7 | 15.21 | 1 | 15.21 | 45.99 | 0.00 |
| CFG1 | 112.25 | 1 | 112.25 | 268.27 | 0.00 |
| CFG2 | 62.28 | 1 | 62.28 | 144.01 | 0.00 |
| CFG3 | 86.17 | 1 | 86.17 | 175.17 | 0.00 |
| CFG4 | 94.08 | 1 | 94.08 | 188.88 | 0.00 |
| CFG5 | 55.77 | 1 | 55.77 | 143.11 | 0.00 |
| CFG6 | 90.01 | 1 | 90.01 | 238.22 | 0.00 |

Note: [1], gl corresponds to degrees of freedom; [2], Fisher statistical value.

### 4.1. Cluster One: Moderate

This cluster was composed of 109 men (48.7%) and 115 women (51.3%), of whom 18.8% stated that they were between 18 and 25 years old, 34.8% between 26 and 34 years old, 38.4% between 35 and 49 years old, 7.6% between 50 and 58 years old and 0.4% over 60 years old. In turn, 54.9% reported having an educational level equal to or higher than professional, distributed as follows: undergraduate 5.8%, bachelor's degree 39.3%, professional and advance degree 50%, and master's degree 89.9%. About the number of children, 45.1% said they did not have any, 27.7% had one, 20.5% had two, and 6.7% had three or more. Compared to their occupation, 21% said they were independent, 60.7% wage-earning, 12.5% students, 1.3 pensioners, 3.1% unemployed, and 1.3% in charge of domestic work. The average monthly income of this conglomerate was characterized by low average, so 57.1% had income between less than one current minimum monthly wage (SMMLV) to two SMMLV, 26.4% between two SMMLV to five SMMLV, and 16.5% more than five SMMLV.

However, as far as the quality of life behaviours of this conglomerate are concerned, it became evident that individuals were seeking to reduce the consumption of goods and services, as well as their waste. Therefore, 92.9% stated that they strive to reduce the misuse of products (e.g., turning off the light, fan, or air conditioning when not in use), in addition to avoiding excessive consumption of these products (89.9%), the attempts to reduce overuse of products are also reinforced in habits such as not ordering more food than can be consumed (84.4%) or beverages (91.1%). As opposed to the reuse of products, 67.9% gave second uses to everyday items such as newspapers, paper (89.8%), clothes (87.5%), or plastic bags. Finally, 82.2% of consumer clusters plan their purchases before making them, in order not to acquire more products than needed.

In the case of behaviour linked to environmental care, 92% of consumers belonging to this cluster were concerned about the ecosystem. However, the proportion of individuals in this cluster who used ecological products was 25.25% less than those who reported being concerned about the environment. Thus, 65.9% of consumers in this cluster usually buy products in biodegradable packaging, while 63.9% claimed to use ecological goods or services, while 69.6% were in the habit of buying or using products which are considered to be environmentally friendly. On the other hand, 65.6% were not willing to spend extra money on eco-friendly products.

With relation to resources for future generations, 75.9% stated that they were aware that the excessive consumption of goods and services at present will result in the impos-

sibility of satisfying the basic needs of the population in the future, as well as the consequences of them on the quality of life for future generations (69.1%) and the widespread concern about the scarcity of natural resources (81.8%). Moreover, 84.4% reported that they were concerned about satisfying basic needs such as access to water, and 79.5% said they minimized their consumption for the sake of preserving ecosystem systems and with them their resources.

Regarding the correlation analysis, it was determined that for this cluster, the item "I care about the environment" was highly related to: "I love our planet" (r = 0.70, p-value = 0.00), "I care about the scarcity of natural resources" (r = 0.69, p-value = 0.00), "I buy and use products that are environmentally friendly" (r = 0.56, p-value = 0. 00) and "I care about meeting the needs of the next generation (e.g., access to water)" (r = 0.50, p-value = 0.00). In summary, and as shown in Figure 4, consumption behaviours linked to resources for future generations are highly correlated between these, however, no high relationships between the other types of behaviour were evident. Finally, for this conglomerate, no correlations were established between the sociodemographic variables and the behaviours analysed.

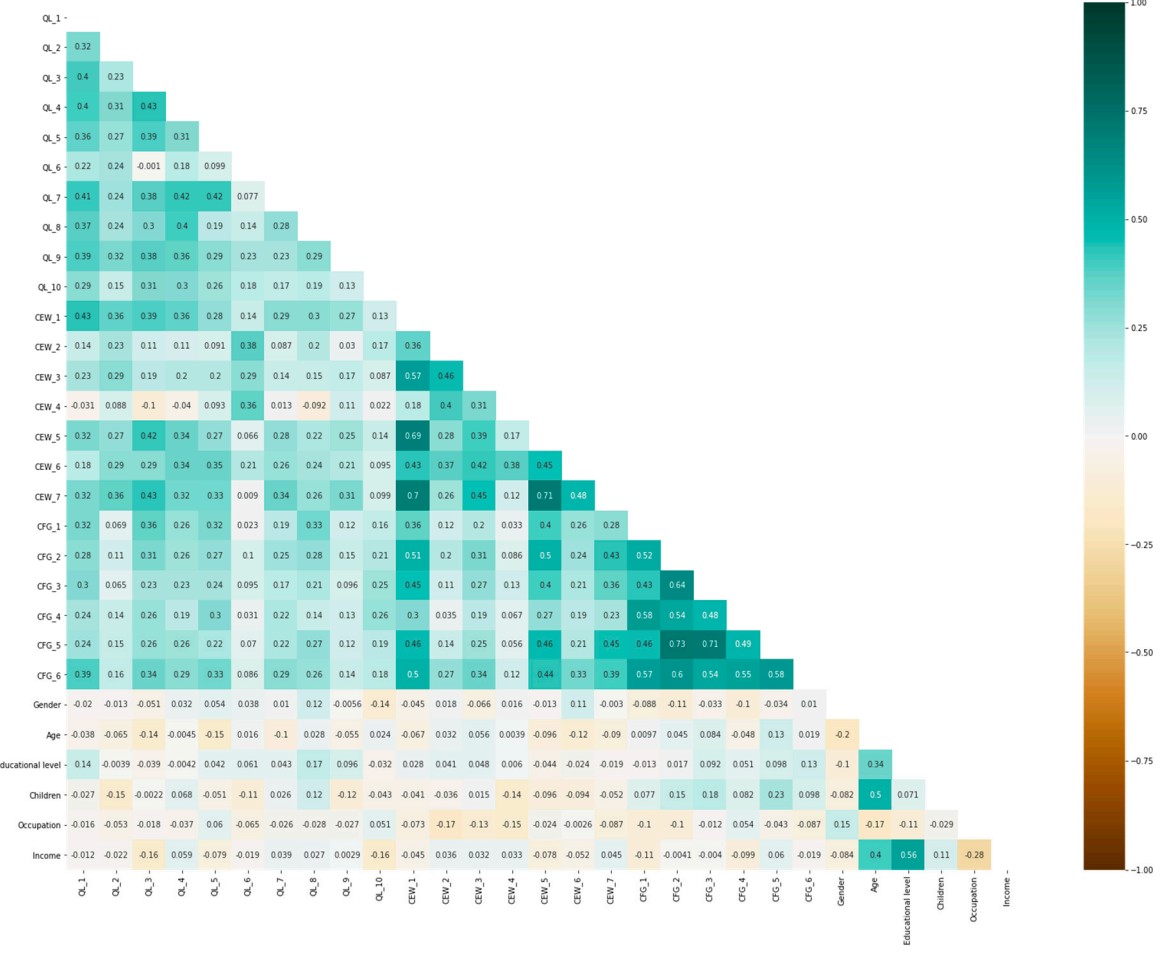

**Figure 4.** Triangular heat map of sustainable consumption behaviour and socio-demographic variables of cluster one. Note: The measurement scale is expressed between −1 to 1, so dark green values represent high positive correlations, otherwise those in dark yellow represent high negative correlations.

### 4.2. Cluster Two: Sustainable

This conglomerate was made up of 77 men (45.5%) and 92 women (54.5%), of whom 10.1% reported being between 18 and 25 years old, 32% between 26 and 34 years old, 47.9% between 35 and 49 years old, 8.9% between 50 and 59 years old and over 60 years old, 1.1%. Compared to the educational level, 48.8% of consumers belonging to this cluster

reported having an educational level equal to or higher than the professional level, distributed as follows: undergraduate 7.1%, bachelor's degree 34.3%, professional and advanced degree 44.4%, master's degree 13.6% and doctorate 0.6%. Regarding the number of children, 32% reported not having any, 24.9% had one, 26% had two, and 17.1% had more than three. In the case of their occupation, 18.3% said they were self-employed, 57.4% were salaried, 14.8% were students, 1.2% were pensioners, 6.5% were students, and 1.8% were in charge of domestic work. On the other hand, 66% had income between one SMMLV and two SMMLV, 26.3% between two SMMLV to five SMMLV and 7.7% more than five SMMLV.

However, with reference to the quality of life behaviours of this conglomerate, it was determined that they are in constant search of reducing their consumption, as well as the waste of goods and services. Therefore, all individuals stated that they were seeking to reduce the misuse of goods and services, in turn, 99.4% avoided excessive consumption of products, which is represented in habits such as not ordering more food than they can consume (99.4%) or wasting drinks or food (100%). In the case of practices related to the reuse of goods or services, 90% gave secondary uses to everyday products such as newspapers, paper (99.4%), clothes (99.4%), or plastic bags (99.4%). With regards to planning purchases, 94.7% said they carried out this activity.

In the case of behaviours related to the care of the environment, 98.88% of the individuals of whom the group is composed expressed concern for the ecosystems and their balance. In this sense, 93.5% of consumers reported using ecological goods or services, 90.2% bought or used eco-friendly products, 91.1% said they bought products with biodegradable packaging and 90.1% preferred to use paper bags as they are biodegradable. Finally, 71% were willing to pay extra money on products considered to be eco-friendly.

In terms of resource behaviour for future generations, 99.4% of consumers in this cluster showed concern that their excessive consumption could affect the well-being of future generations and 96.4% are concerned about the scarcity of natural resources. In turn, 98.2% said they were concerned about the satisfaction of future basic needs such as access to water. In addition, 98.2% are trying to control their desire for purchases.

With respect to the correlation analysis for this cluster, it was evident that there are no high relationships between sustainable consumption behaviours, as shown in Figure 5, however, average correlations were established for the item "I buy and use products that are environmentally friendly" and "I choose to buy products with biodegradable packaging or packing" ($r = 0.42$, *p*-value = 0.00) and "I use environmentally friendly products and services" ($r = 0.49$, *p*-value = 0.00), as well as "I often think about the quality of life of future generations" and "I care about the future generation" ($r = 0.41$, *p*-value = 0.00).

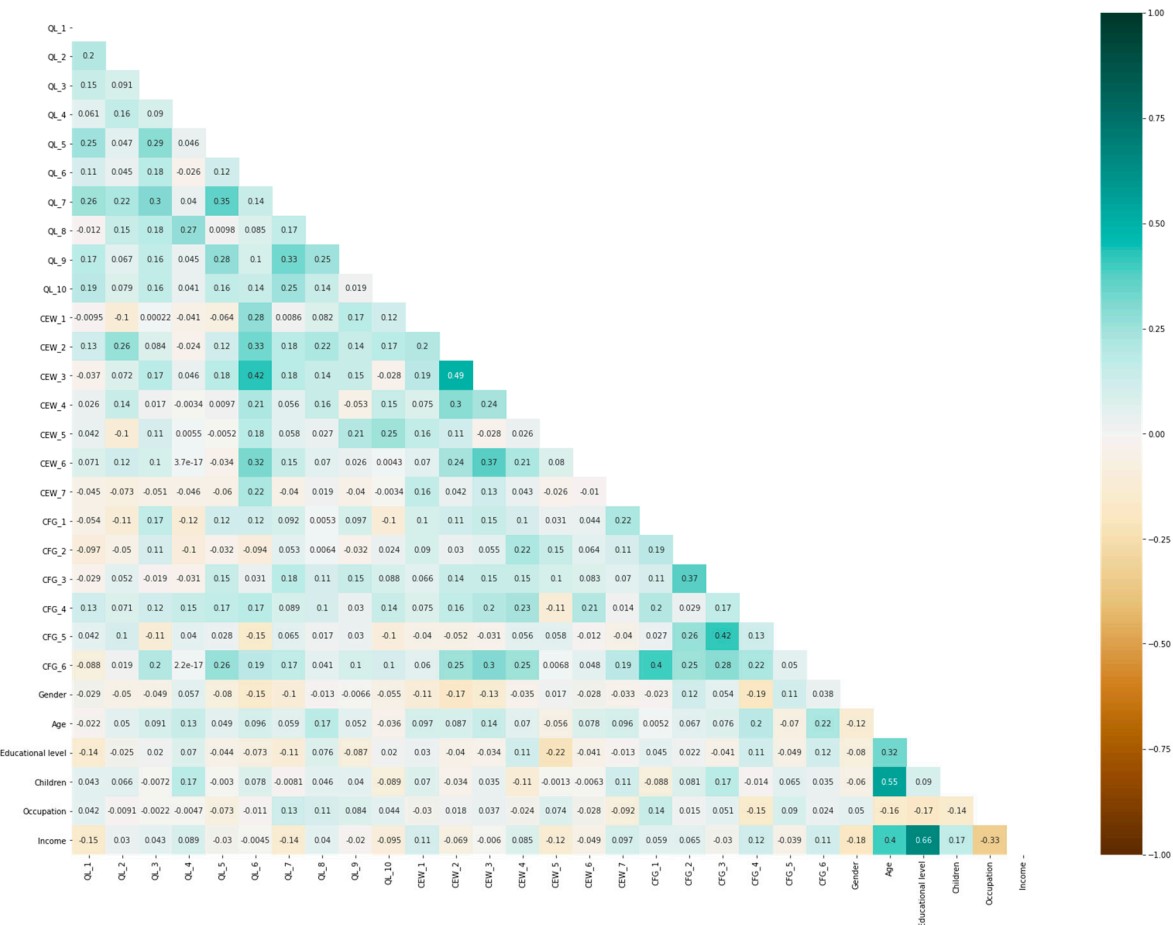

**Figure 5.** Triangular heat map of sustainable consumption behaviour and socio-demographic variables of cluster two. Note: The measurement scale is expressed between −1 to 1, so dark green values represent high positive correlations, otherwise those in dark yellow represent high negative correlations.

## 5. Discussion

As determined in the results section, low-impact sustainable consumption behaviours have been appropriated in various ways by Colombian consumers. This is evident both in the EFA and in the hierarchical analysis of the clusters. Thus, regarding the EFA, it was determined that the sustainable consumption behaviours evaluated were grouped in a similar way to the proposal of Quoquab et al. [17], only for the items CEW5, CEW7, and QL6 were they charged to different factors than those raised by the original instrument. So, in front of the national scenario CEW5 is part of the resource behaviours for future generations, CEW7 of quality of life and QL6 of environmental care.

However, Colombian consumers were grouped into two clusters. Thus, it is recognized that for both cases, in terms of quality of life, behaviours such as consumption reduction, product reuse, waste avoidance, and purchase planning are manifested. These behaviours are usually influenced by the degree of concern that the individual has about his consumption in relation to the effects on the environment [73,74], or by economic variables that, from the theory of consumption efficiency, individuals seek to meet needs without incurring higher costs, represented by lower consumption, the development of new functions to the product, among others [75]. Nevertheless, for the case of both conglomerates, no high correlations were determined between the items QL1, QL2, QL3, QL4, QL5, QL6, QL7, QL8, QL9, and QL10 with CEW1, CEW5 CEW7, CFG3, CFG5, or the level of economic income. For that reason, the adoption of such behaviours derives from the

influence of other variables that are not related in this study, such as environmental education [19], previous habits developed together with the family or social circle [46,51], among others.

In the case of environmental care behaviour, consumers generally expressed concern about the current state of ecosystems and the scarcity of natural resources. Thus, for cluster one, it was determined that these habits have not been fully appropriated by those individuals who were included in the group, which marks some differences with cluster two. In other words, it can be said that people belonging to cluster one are less likely to use ecological products or those considered eco-friendly, as well as being reluctant to pay extra money for the types of products described. This is in line with the findings of Chan [76] and Welsch and Kühling [40], who recognised in other consumption scenarios that, although people may be concerned about the impact of their consumption on the environment, for economic reasons they do not usually purchase goods or services with eco-friendly characteristics, especially those related to organic food. However, in this conglomerate, there were high correlations between levels of concern (CEW1) and the purchase of environmentally friendly products (CEW3), in line with Ajzen [73] who highlights that the greater the degree of uncertainty on the part of consumers about the environmental future, the more willing they will be to buy products with ecological characteristics.

In terms of resource-related behaviour for future generations, the second cluster tends to show higher levels of concern about over-consumption in terms of the inability to meet the basic needs of the population, and its impact on quality of life. However, it is highlighted in cluster one that, as with environmental care behaviours, there is a high correlation between their concern about the scarcity of resources and the satisfaction of future needs, resulting in a possible reinforcement of environmental care habits.

Considering what has been previously stated in the holistic analysis proposed in this paper, it was possible to determine in a general way that Colombian consumers appropriate the habits in different ways as shown in the comparison of the three main categories of low-impact sustainable consumption behaviour. This is in line with the theory expressed by [17,22], which indicates that a high level of appropriation in one of these behaviours does not result in a high level of appropriation in the other two, evidenced by the few high correlations between the items evaluated for both clusters. This can derive from the level of consumer commitment, expressed in environmental attitude, perceived responsibility, behavioural efficiency, among others [51]. Although these factors were not analysed in the present work, which is limited to behaviour, it is recognised that they are present in the population because of the existence of a positive attitude towards environmental issues, which is evaluated in the item (CEW 7), or because of their sense of responsibility towards society. On the other hand, the results on resource behaviour for future generations presented in this work enrich the literature in this category by indicating that the population has acquired various habits to minimise its consumption and to save these resources for the future, although the limitation of the items evaluated by the scale is recognised by not going into more detail on behaviour such as the work developed by Bulut et al. [62] or O'Rourke et al. [65].

In view of the implications of the results presented for Colombia, it is necessary to recognise that public policymakers, in terms of sustainable consumption, must dissociate themselves from the traditional view that assumes that consumers have not appropriated low-impact behaviours related to this type of consumption, as observed in the country's National Policy on Sustainable Production and Consumption [13]. For this reason, policies aimed at reinforcing these habits in the population must be enhanced in order to stimulate the appropriation of those habits known to have a high impact and to focus the efforts on the companies so that they develop sustainable processes that, along with the habits acquired by consumers, grant the development of a sustainable model that can effectively break the paradigms linked to economic growth. On the other hand, for marketing spe-

cialists, and in congruence with the appropriation of habits previously described, products should continue to be developed the recycling and reuse characteristic or generate strategies of massification of existing eco-friendly products by changing the price strategy and permitting the consumers to reinforce caring of the environment behaviours.

Finally, the results should be analysed from the limitations of the study taking into consideration the following: 1) the conceptualization of the study is exploratory and inconclusive and the existing transversality in the collection of data does not concede the evaluation of changes in behaviour over a period of time as the main critics of this type of study highlight [32]; 2) the size of the sample can determine through another type of analysis (e.g., Confirmatory factor analysis); 3) the socio-demographic characteristics of the population that do not allow for particularised analyses on more specific groups of individuals, given that the sample does not have a similar distribution between age ranges, educational level, number of children, among others; 4) the lack of discrimination between the urban and the rural or the limitations of the scale proposed by Quoquab et al. [17] who incorporated attitudinal components into this scale; 5) the lack of specificity of some of the behaviours analysed in this self-reporting scale.

Recognizing that Colombian consumers should continue to develop studies that evaluate their behaviour and other factors expressed in this discussion, such as the incidence of economic variables (e.g., the cost of products, speculation in the prices of environmentally friendly products), beliefs and opinions related to this type of behaviour, and attitudes that gran the development of these behaviours, among others. Additionally, it is necessary to consider how high-impact sustainable consumption behaviours are approached in countries with a high degree of inequality since it cannot be ignored that a part of the population contributes to the development of these attitudes and behaviours.

## 6. Conclusions

Several conclusions were reached in this study, the aim of which was to explore low-impact sustainable consumption behaviour in Colombia and the convergence and divergence of this type of consumer behaviour in the country. The study showed that the population has appropriated, to a greater or lesser extent the habits, related to this type of consumption, especially in terms of quality of life and resources for future generations. Otherwise, it was determined that in those behaviours related to the care of the environment, one of the main barriers in their adoption is the economic factor, in which a part of the population does not usually invest money in ecosystem-friendly products due to the high costs they represent.

On the other hand, no correlation between low-impact consumption behaviours was generally established for cluster two. Thus, only for cluster one was this situation present in some of the behaviours of environmental care and resources for future generations, so there is no positive or negative reinforcement between them. Moreover, for both clusters, the sociodemographic variables were not related to the behaviours, so that, in the present exploratory study, variables such as sex, age, educational level, number of children, occupation or income level have no incidence on the behaviours analysed.

Taking into consideration, the limitations of this study—such as the size of the sample, the transversality, the lack of discrimination between urban and rural areas, and the limitations of scale applied due to self-reporting—it can be stated that Colombians, through their behaviour, tend to care for natural systems, and in this way contribute to the mitigation of climate change. Based on this, we must continue to manage the appropriation by consumers through the development of public policies that recognize that these behaviours are implicit in society, in order to initiate a process of adoption of the so-called high-impact behaviours, through the development of economic stimuli for the population, and not only for companies. With this, it will be possible to comply with what has been agreed by the national government in the various international treaties, especially Sustainable Development Goal 12. In parallel, other decision-makers, such as marketing specialists, must continue to promote this type of consumption among the population

through corporate social responsibility campaigns, and the development of new eco-friendly products that are more affordable to the population in economic terms.

**Author Contributions:** Conceptualization, A.G.R. and R.L.C.B.; methodology, A.G.R., B.R.-C and E.M-C.Á.; software, A.G.R.; validation, R.L.C.B., and B.R.-C.; formal analysis, A.G.R.; investigation, A.G.R., R.L.C.B., E.M-C.Á., B.R.-C.; resources, A.G.R.; data curation, A.G.R.; writing—original draft preparation, A.G and R.L.C.B.; writing—review and editing, A.G.R., R.L.C.B., E.M-C.Á. and B.R.-C.; project administration, A.G.R.; All authors have read and agreed to the published version of the manuscript.

**Funding:** This research was funded by Corporación Universitaria de Asturias 001-2020.

**Institutional Review Board Statement:** The study was conducted according to the guidelines of the Declaration of Helsinki, and approved by the Institutional Review Board of University Corporation of Asturias (protocol code 02-2020 and date of approval 11 January 2020).

**Informed Consent Statement:** Informed consent was obtained from all subjects involved in the study.

**Data Availability Statement:** The data presented in this study are available on request from the corresponding author. The data are not publicly available due to the current Colombian laws that require the signing of a data transfer contract between the Corporation of Asturias and the applicants.

**Acknowledgments:** To Cecilia Carabajal who, with her unconditional support, made the style correction and translation of this article.

**Conflicts of Interest:** The authors declare no conflict of interest.

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
