# Peer review of "Sustainable Consumption Behaviour in Colombia: An Exploratory Analysis"

_sustainability, doi:10.3390/su13020802_

Round 1

Reviewer 1 Report

Dear authors,

Your research approaches an ardent socio-economic and environmental topic, focused on the case of Colombia.

I suggest improving the quality of the submitted paper based on the attached PDF document that contains 72 comments.

Besides those 72 comments linked to specific areas in the PDF file, please also consider the following:

  1. The Discussion section should be extended. The discussion should be done in the light of explaining the connection between the three types of sustainable consumption behaviours and the manners suggested by the authors for fostering sustainable consumption behaviors, based on the motivations of each sustainable consumption behaviour identified (referring to the clusters).
  2. Please add a paragraph about the novelty factor of your paper.
  3. In the Conclusions section, please consider explaining the main limitations of this study and suggest further research directions.

Best regards,

Reviewer

Author Response

Kind regards, dear reviewer,

We are sending you the responses to each of your comments and how they were incorporated into the body of the document.

Point 1: The Discussion section should be extended. The discussion should be done in the light of explaining the connection between the three types of sustainable consumption behaviours and the manners suggested by the authors for fostering sustainable consumption behaviors, based on the motivations of each sustainable consumption behaviour identified (referring to the clusters).

Response 1: This point has been adjusted in the discussion of the document by showing how the types of sustainable consumption behaviour are related from a holistic analysis stated in  the introduction of the document. Additionally, both causes and reasons for this type of behaviour are considered in the clusters which were not analysed in the document but are useful to explain up to some extent the appropriation of this type of consumer habits.

Point 2: Please add a paragraph about the novelty factor of your paper.

Response 2: A paragraph dealing with the novelty of the study from the theoretical perspective of a holistic view was added both in the introduction and in the discussion. In this case, the originality of this topic in the academic field and in the study developed for the country are clarified.

Point 3: In the Conclusions section, please consider explaining the main limitations of this study and suggest further research directions.

Response 3: In accordance with the format required by the journal and the specifications given in the template, the limitations and future strands of research were added in the discussion section.  Additionally, a synthesis of some of these limitations and strand of research was made in the conclusions.

Point 4: I suggest improving the quality of the submitted paper based on the attached PDF document that contains 72 comments.

Response 4: The 72 comments are answered in the manuscript as they are considered to be correct for the development of the document. All the points in this reply are not specified because of the length that it would represent; however, the responses will be clearly perceived in the reading,

Reviewer 2 Report

I consider the research topic to be interesting and topical. But in my humble opinion, I also consider that research does not have enough novelty and the results are not relevant enough to be published in a journal of the academic level of Sustainability, with high impact factor.

Some comments for improvement:

  • There are, especially in the introduction, some confusing and difficult to read sentences.
  • It should be improved the purpose of research, because it is too vague and does not clarify the goal of empirical research (This paper aims to explore low-impact sustainable consumption Behavior in Colombia and the relationship between them).
  • The introduction should not have as many sections or sub-sections. A specific section should be created to serve as the theoretical framework or review of the previous literature on sustainable consumption.
  • In the introduction section it is not clearly indicated that the authors will use the classification proposed by Quoquab et al. (2017).
  • The classification of behaviors that the authors propose is not very clarifying. As can be deduced from reading the methodology section, it is based on the study by Quoquab et al. (2017), but it is not a 100% adequate scale to measure behaviors. There are items that measure concerns and attitudes, not behaviors (real actions).
  • About the methodology. The authors should indicate if the sample is representative of the national population of Colombia according to the different sociodemographic criteria.
  • I believe that the original scale of Quoquab et al. (2017) is 24 items, but the authors only use 23.
  • Methodologically, I believe that the main problem of the research is the use of the classification of the items obtained in the study by Quoquab et al. (2017). But a prior factor analysis should be done to identify what factors are obtained in the case of the sample of the Colombian population. Perhaps the three dimensions of the original study are not obtained, or the grouping of the items is not the same.
  • Keywords: they must be words that facilitate the identification of the topic and the search in article search engines. It is not suitable to use “sustainable consumption” and “sustainable consumption behavior” at the same time. Nor is it appropriate to use “well-being or quality of life”, as it should be divided into two keywords: “well-being” and “quality of life”.
  • Authors say: “Consequently, the lack of knowledge about sustainable consumption behaviour in Colombia hinders the work of public policy makers, marketing specialists and other decision-makers in terms of sustainable development, and an analysis of this type of behaviour is required in order to legitimise the strategies, policies, regulations and decisions taken”.  However, there is neither a section nor a few paragraphs dedicated to discussing the managerial implications of the results obtained. That is, what is the value of these results for decision makers in the administration and in companies?

Author Response

Dear reviewer,

We are sending you the responses to each of your comments and how they were incorporated into the body of the document.

Kind regards.

Point 1: There are, especially in the introduction, some confusing and difficult to read sentences.

Response 1: The document was corrected and revised in order to eliminate any difficulties and errors of interpretation in the introduction and in the whole manuscrit in general.

Point 2: It should be improved the purpose of research, because it is too vague and does not clarify the goal of empirical research (This paper aims to explore low-impact sustainable consumption Behavior in Colombia and the relationship between them).

Response 2: The objective of the research was adjusted in the summary as well as in the introduction and the conclusion.

Point 3: The introduction should not have as many sections or sub-sections. A specific section should be created to serve as the theoretical framework or review of the previous literature on sustainable consumption.

Response 3: The enumeration of the document was adjusted with the sole objective of not presenting an introduction with many sections.

Point 4: In the introduction section it is not clearly indicated that the authors will use the classification proposed by Quoquab et al. (2017).

Response 4: This aspect has been adjusted in the document, specifically in the introduction.

Point 5: The classification of behaviors that the authors propose is not very clarifying. As can be deduced from reading the methodology section, it is based on the study by Quoquab et al. (2017), but it is not a 100% adequate scale to measure behaviors. There are items that measure concerns and attitudes, not behaviors (real actions).

Response 5: The adjustment on the classification of sustainable consumption behaviours was made as shown in section 2.1. It is important to highlight the holistic view of these behaviours which are presented.

Point 6: About the methodology. The authors should indicate if the sample is representative of the national population of Colombia according to the different sociodemographic criteria.

Response 6: In the original document this assertion was not made. However, the confusion that could arise in the text is understandable; for that reason the relevance of the sample for an exploratory study is justified and a simile study is carried out with resembling samples on populations of national character.

Point 7: Methodologically, I believe that the main problem of the research is the use of the classification of the items obtained in the study by Quoquab et al. (2017). But a prior factor analysis should be done to identify what factors are obtained in the case of the sample of the Colombian population. Perhaps the three dimensions of the original study are not obtained, or the grouping of the items is not the same.

Response 7: As authors, we understand your point of view; however, we recognize that in order to apply a factorial analysis, larger samples are required. This is also specified as a limitation in the development of the study

Point 7: Keywords: they must be words that facilitate the identification of the topic and the search in article search engines. It is not suitable to use “sustainable consumption” and “sustainable consumption behavior” at the same time. Nor is it appropriate to use “well-being or quality of life”, as it should be divided into two keywords: “well-being” and “quality of life”.

Response 7: Based on the suggestion of other evaluation pairs, the keyword section was revised.

Point 8: Authors say: “Consequently, the lack of knowledge about sustainable consumption behaviour in Colombia hinders the work of public policy makers, marketing specialists and other decision-makers in terms of sustainable development, and an analysis of this type of behaviour is required in order to legitimise the strategies, policies, regulations and decisions taken”.  However, there is neither a section nor a few paragraphs dedicated to discussing the managerial implications of the results obtained. That is, what is the value of these results for decision makers in the administration and in companies?

Response 8: We considered that the value of these results was detailed in the conclusions; however, we proceeded to expand on this point in the discussion as suggested.

Reviewer 3 Report

The paper need some improvements to increase the quality of the research. First, I understand that there are no results from country like  Columbia regarding these concepts BUT I think that isn't apropiated to mention  results from well developed countries like Belgium and Germany for comparison. Some suggestions will found bellow:

  1. In the entire paper are the same expression "well-being OR quality of life". These are two concepts totally different. Also, to the end of the paper "well-being" is replace with "welfare" that a still a different concept. there are no sinonimouse!!
  2. In the ABSTRACT, I don't understand the sentence (line 16) "Accordingly, the aim of this work is to explore the low-impact sustainable consumption behaviour in Colombia and the relationship between them". Them means....???? This sentence is also in the manuscript, identically.
  3. The authors must reformulated in the abstract (lines 20-21) "With 20 the data collected, we proceeded to identify how consumers are grouped".
  4. Please cut from the keywords the words identically from the title.
  5. The Introduction is too long Please split into 1. Introduction and 2. Literature review/Conceptual framework/Background research (I suggest from the begin the section 2. Literature review from the subsection 1.1. until material and methods)
  6. There are so many citation (and some references) without the year of publication:
    • line 36 for the report,
    • line 48 and reference [10],
    • line 54-55
    • line 66
    • etc.
  7. For the definition line 45-48 is a good definition from a conference?? Without any concrete author?
  8. Also, line 49, "many conferences"????? Please reformulate!
  9. In the paragraph from line 57 to the line 64, the number 4 is missing!
  10. Line 73, it is the citation in brackets [13] BUT in put also the citation in the APA style. Please correct!
  11. line 79 "in the case of country"??? What country? And line 90.
  12. From the line 103 could begin the section 2. Literature review/theoretical framework
  13. line 118, in the brackets it put [e.g....]. Please correct also at line 162.
  14. Subtitle 1.1.1. it isn't a proper title, please reformulate.
  15. Comparisons with Germany and Belgium isn't equal, for Columbia!!!
  16. line 153 "is facilitated since it facilitates". Please reformulate.
  17. All the tables with examples have not sources. there are made by the authors using some references???Must mentioned the sources!
  18. line 161, the are some duplicate for source [17].
  19. The phrase from line 163 to line 166 is too long.
  20. line 196-197: the authors used the quota technique (ratio of 1 to 3) NOT the snowball one.Please correct. The snowball is in fact a sampling techniques totally different from the quota one.
  21. Regarding the education level for demographic criteria: please used the international structure :bachelor, undergraduate, university degree, etc NOT the national one!
  22. Lines 207-208, it must mention the characteristic, respectively the occupational status.
  23. line 218: the method was DEVELOPED or applied?
  24. In the section 3. Material and methods, there are no mention regrading the statistical soft used for the methods: SPSS, Excel, etc????Please complete with this information.
  25. Using a Likert scale, I don't find the results for Cronbach alpha!!
  26. Lines 234-235: o correltiaon coefficient about 0.5 is a HIGH one???? from statistical point of you IS A MEDIUM?MODERATE ONE!!! High is over 0.75!!
  27. line 237 "compare" OR analysing the results ???
  28. Figure 2, please put the legend: 1 is for cluster? and 2 for cluster?
  29. For data from Table 5, the normal distribution for these data was tested???
  30. Lines 261 - 262....making the amount there are more than 100%, please correct!
  31. Line 269 the well-being concept is replace with "welfare"!!!Lines 301-302: please give some explanations for this situations!!
  32. Please move the Figure 4 near to the text NOT into Disscussions!

Author Response

Dear reviewer,

We are sending you the responses to each of your comments and how they were incorporated into the body of the document.

Kind regards.

Point 1: The paper needs some improvements to increase the quality of the research. First, I understand that there are no results from countries like  Colombia regarding these concepts BUT I think that isn't appropriate to mention  results from well developed countries like Belgium and Germany for comparison.

Response 1: Due to the gap existing in the literature for the country, it is necessary to seek experience from other countries in order to be able to compare the results of the study in the discussion and to observe whether there are similar or different behaviours of the population with other countries. The purpose of referencing this study is not only to understand the differences and limitations but also to acknowledge that advances have been made in the analysis of this type of behaviour.

Point 2: In the entire paper are the same expression "well-being OR quality of life". These are two totally different concepts. Also, to the end of the paper "well-being" is replaced with "welfare" that is still a different concept. there are no sinonimouse!!

Response 2: After reviewing the concepts, we agreed with their point of view, so we proceeded to make the adjustment throughout the document to Quality of Life.

Point 3: In the ABSTRACT, I don't understand the sentence (line 16) "Accordingly, the aim of this work is to explore the low-impact sustainable consumption behaviour in Colombia and the relationship between them". Them means....???? This sentence is also in the manuscript, identically.

Response 3: The adjustment was made after the request of another pair who suggested to describe the objective more broadly.

Point 4: The Introduction is too long Please split into 1. Introduction and 2. Literature review/Conceptual framework/Background research (I suggest from the beginning the section 2. Literature review from the subsection 1.1. until material and methods)

Response 4: This aspect was adjusted in the document. A division was made in the introduction initially proposed in accordance with your respectful request.

Point 5: There are so many citation (and some references) without the year of publication:

Response 5: The document's references were reviewed according to the instruction for the authors and the ACS guide https://pubs.acs.org/doi/pdf/10.1021/bk-2006-STYG.ch014

Point 6: For the definition line 45-48 is a good definition from a conference?? Without any concrete author?

Response 6: The definition found in the text is the result of an international symposium developed by the United Nations and the leaders of the nations regarding sustainable consumption. Hence the representativeness of this concept is considered because of the formal definition adopted by this renowned international body.

Point 7: line 196-197: the authors used the quota technique (ratio of 1 to 3) NOT the snowball one.Please correct. The snowball is in fact a sampling technique totally different from the quota one.

Response 7: The review and adjustment on the type of sampling was made.

Point 7: In section 3. Material and methods, there is no mention regarding the statistical soft used for the methods: SPSS, Excel, etc????Please complete with this information.

Response 7: The software used for the development of the methods has been added.

Point 8: Using a Likert scale, I don't find the results for Cronbach alpha!!

Response 8: This statistic had been omitted. However, as we understand its relevance to comprehend the reliability of the instrument in the Colombian population, it was added with its respective analysis.

Point 8: Lines 234-235: o correltiaon coefficient about 0.5 is a HIGH one???? from statistical point of you IS A MEDIUM?MODERATE ONE!!! High is over 0.75!!

Response 8: Your point of view related to “A correlation greater than 0.8 is generally described as strong, whereas a correlation less than 0.5 is generally described as weak. These values can vary based upon the "type" of data being examined” (Samuel and Ethelbert, 2015), is understood. However, it is recognized by the previous authors mentioned along with Cohen, also quoted in the text, that these values are not relevant in the social sciences, where the perceptions about a dependent variable changes constantly with respect to an independent variable. Hence the correlations against scales derived from biology, physics, engineering, among others, are lower, but against social science studies are considered high.

Point 8: Others points.

Response 8: Due to the number of points related to grammar or changes in form, they were adopted throughout the document and no details were given in this answer because of the length it would represent.

Round 2

Reviewer 2 Report

Despite the changes introduced by the authors in the document, I continue to insist that the weakness of the research is its little novelty and its low contribution to academic knowledge.

Author Response

Point 1: Despite the changes introduced by the authors in the document, I continue to insist that the weakness of the research is its little novelty and its low contribution to academic knowledge.

Response 1: In response to this aspect, the novelty of the research was approached from two perspectives. The first was a holistic analysis of sustainable consumption behaviour, which confirmed a highly theoretical perspective introduced in lines 98 to 109 and discussed in lines 455 to 470; and the second lies in the marginal implications for the development of Colombian public policies, decision-making by marketing specialists, among others.

In addition, and taking into account the review of another pair, a factorial analysis was included in the revised manuscript in order to reinforce the holistic analysis presented in the development of the work.

Reviewer 3 Report

Congrats!

Author Response

Cordial greetings.

I would like to thank you for your comments, we know that they improved our work. We hope to have a positive outcome for the public.

Alfredo Guzmán Rincón

Round 3

Reviewer 2 Report

Current version of the document has improved considerably